# Optimal Sizing of Rooftop Rainwater Harvesting Tanks for Sustainable Domestic Water Use in the West Bank, Palestine

Sameer Shadeed *[ID] and Sandy Alawna

Water and Environmental Studies Institute, An-Najah National University, Nablus P.O. Box 7, Palestine; sandy–alawna@hotmail.com
* Correspondence: sshadeed@najah.edu

**Abstract:** In highly water-poor areas, rooftop rainwater harvesting (RRWH) can be used for a self-sustaining and self-reliant domestic water supply. The designing of an optimal RRWH storage tank is a key parameter to implement a reliable RRWH system. In this study, the optimal size of RRWH storage tanks in the different West Bank governorates was estimated based on monthly (all governorates) and daily (i.e., Nablus) inflow (RRWH) and outflow (domestic water demand, DWD) data. In the estimation of RRWH, five rooftop areas varying between 100 m$^2$ and 300 m$^2$ were selected. Moreover, the reliability of the adopting RRWH system in the different West Bank governorates was tested. Two-time series scenarios were assumed: Scenario 1, S1 (12 months, annual) and scenario 2, S2 (8 months, rainy). As a result, reliable curves for preliminary estimation of optimal RRWH storage tanks for the different West Bank governorates were obtained. Results show that the required storage tank for S1 (annual) is more than that of the S2 (rainy) one. The required storage tank to fulfill DWD is based on the average rooftop area of 150 m$^2$, the average family members of 4.8, and the average DWD of 90 L per capita per day (L/c/d) varies between (75 m$^3$ to 136 m$^3$) and (24 m$^3$ to 84 m$^3$) for S2 for the different West Bank governorates. Further, it is found that the optimal RRWH tank size for the 150 m$^2$ rooftop ranges between 20 m$^3$ (in Jericho) to 75 m$^3$ (in Salfit and Nablus) and between 20 m$^3$ (in Jericho) to 51 m$^3$ (in Jerusalem) for S1 and S2 scenarios, respectively. Finally, results show that the implementation of an RRWH system for a rooftop area of 150 m$^2$ and family members of 4.8 is reliable for all of the West Bank governorates except Jericho. Whereas, the reliability doesn't exceed 19% for the two scenarios. However, the reduction of DWDv is highly affecting the reliability of adopting RRWH systems in Jericho (the least rainfall governorate). For instance, a family DWDv of 3.2 m$^3$/month (25% of the average family DWDv in the West Bank) will increase the reliability at a rooftop area of 150 m$^2$ to 51% and 76% for S1 and S2, respectively.

**Keywords:** rooftop rainwater harvesting; optimal size; domestic water; water poverty; GIS; West Bank

## 1. Introduction

The misuse of different water resources has been leading to an increase of stress on these resources, thus impacting water availability, especially in arid and semi-arid regions [1]. This situation urges the necessity to look into more sustainable water resource options. For instance, rainwater harvesting (RWH) is deemed to be a viable option to satisfy water needs for different purposes, among which, domestic use is the most important [2,3].

Worldwide, domestic water needs represent about 10% of total water demand [4]. Thus, in arid and semi-arid areas, rooftop rainwater harvesting (RRWH) can be used as a sustainable option to overcome domestic water shortages [5–7]. RRWH is a simple technique for totally or partially satisfying domestic water demand (DWD) by collecting and storing rooftops rainwater [8]. The RRWH system consists of three main components; catchment area (rooftop), conveyance system, and storage tank [9].

In the West Bank, domestic water shortage is a dominant problem, where 57% of the total West Bank area is under high to very high domestic water poverty conditions [10].

However, domestic RWH is highly suitable in about 60% of the total West Bank area [10]. Moreover, the potential average annual RRWH volume in the high water-poor areas is about 32 million cubic meters (MCM) [11].

Potential RRWH volume (inflow) and DWD (outflow) are the key parameters in designing an optimal and reliable RRWH storage tank [12,13]. The storage tank is the most expensive component of an RRWH system [14]. However, the optimal RRWH storage tank is affected by several factors which include: Rainfall depth, rooftop area, availability of alternative water supply, water demand, and socio-economic constraints [9,15–18]. The estimation of an RRWH storage tank can be done by using a daily or monthly water balance approach [19]. To estimate an optimal RRWH tank size with an acceptable accuracy, a one-month time interval is commonly used [20]. In this study, the average monthly values of rainfall and DWD were used in the RRWH tank size estimation.

Given the monthly data of inflow and outflow, the optimal RRWH tank size can be estimated based on different methods. The empirical [17], stochastic parametric and non-parametric (or probabilistic analytical) [21,22], and continuous mass balance (e.g., Rippl) methods [8,23]. Empirical methods depend on developing empirical relations to describe the RRWH tank sizing. Stochastic methods were used to simulate important missing parameters in the tank size estimation of parametric and non-parametric approaches [24]. However, continuous simulation through the mass balance equation is the most popular approach used to estimate RRWH tank size [25].

The estimation of RRWH tank size can be accomplished based either on supply size or demand size, where supply size is the maximum volume of RRWH that can be harvested from a certain rooftop area. Demand size represents the maximum volume of water that is required to satisfy the water needs of a certain family during a specific period [26]. In this study, the mass balance between water supply (from RRWH) and demand (family water consumption) was conducted to estimate the required RRWH tank size. According to Rippl [27], the maximum cumulative positive differences between demand and supply represent the required RRWH tank size.

In the body of literature, there exist several studies that were focused on the estimation of optimal RRWH tank size. For instance, in Jordan [26,28], in Mexico [4], in Greece [19], in Northern Cyprus [29], in the north of Portugal [30], in Taiwan [13], in Iran [31], and in Australia [32,33].

This research aims to estimate optimal RRWH tank size for the different West Bank governorates based on a range of rooftop areas (100 m$^2$ to 300 m$^2$) and given the average DWD of 90 L per capita per day (L/c/d) and average family members of 4.8 [34]. Two scenarios were tested; scenario 1 (S1), which is based on the annual rainfall (12 months), and scenario 2 (S2), which covers the rainy months from October to May (8 months).

The added value of this research is the development (for the first ever-time) of a reliable relation between rooftop areas and optimal RRWH tank size based on the average DWD in the West Bank. The novelty of this research can be summarized by the prediction of reliable RRWH storage tanks in the different West Bank governorates, which in turn can guide decision-makers toward sustainable utilization of potential RRWH, totally or partially, in Palestine. The applied approach can be used, mainly in arid and semi-arid areas, where water supply is uncertain and RRWH is deemed to be a robust domestic water supply.

## 2. Materials and Methods

### 2.1. Study Area

The study area is the West Bank with a total surface area of 5658 Km$^2$ [35]. Administratively, the West Bank is divided into 11 governorates; Salfit, Jenin, Tubas, Nablus, Jericho, Hebron, Tulkarm, Qalqiliya, Jerusalem, Bethlehem, and Ramallah and Al-Bireh (see Figure 1). However, the total population in the West Bank is nearly 2.88 million [36]. The surface elevation of the West Bank ranges between 410 m below mean sea level (msl) in the Dead Sea and 1022 m above (msl) in Tall Asur in Hebron [37]. Mediterranean climate

is prevailing in the West Bank, which is characterized as hot and dry in summer and wet and cold in winter [37]. The West Bank rainy season usually extends over 8 months from October to May and most of the West Bank rainfalls (80%) in winter (December to February) [38,39]. Moreover, the rainfall in the West Bank is characterized by high temporal and spatial variation. The long-term annual average varies between 133 mm in the proximity of the Jordan River (in Jericho) to 658 mm in the central mountains (in Salfit) with an annual average value of about 420 mm for the entire West Bank (see Figure 1).

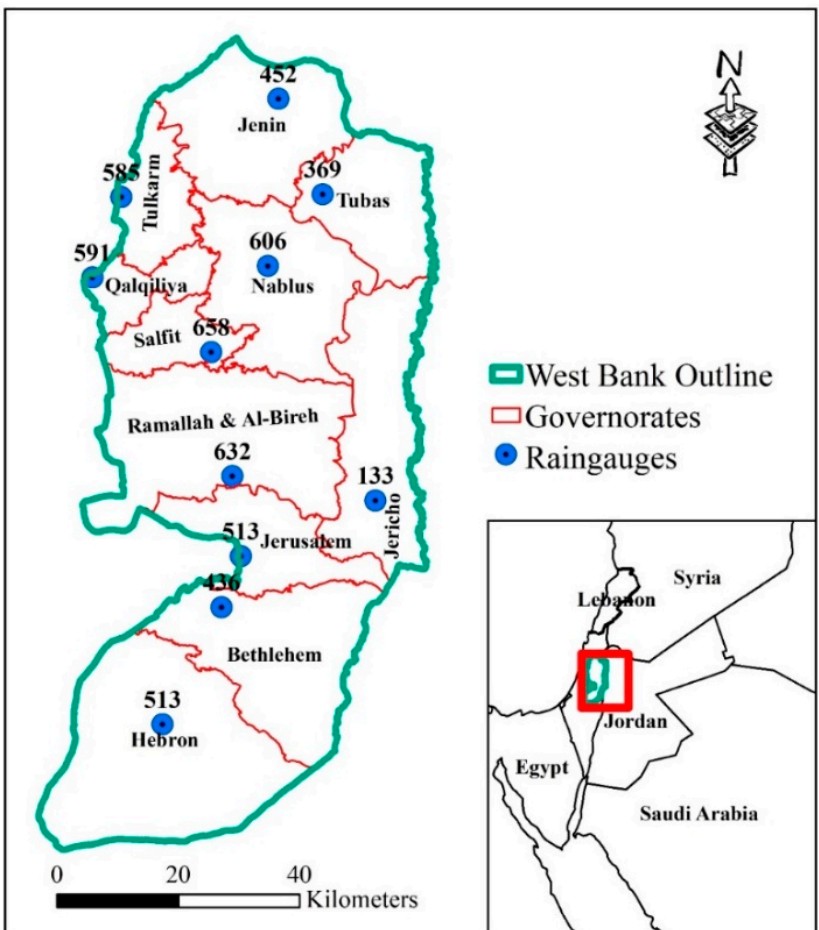

**Figure 1.** The regional location map of the West Bank together with the long-term annual average rainfall (in mm).

The West Bank has been divided into four agro-climatic zones; sub-humid (22%), semi-arid (44%), arid (30%), and hyper-arid (4%) (see Figure 2) [40]. The land use map is classified into seven categories: Arable land (supporting grains), built-up areas, irrigated farming (supporting vegetables), Israeli settlements, permanent crops (grapes, olives, citrus, and other fruit trees), rough grazing/subsistence farming, and woodland/forest. In this research, the focus is on the built-up areas where rooftops account for about 28% that was considered in the estimation of RRWH volumes in the different governorates [35].

In the West Bank, water is being obtained either from groundwater wells and springs or purchased from an Israeli water company (Mekorot). According to the Palestinian water authority reports, the domestic water demand, consumption, and supply are 146 MCM, 88 MCM, and 119 MCM, respectively [34]. Accordingly, the domestic water supply-demand gap is nearly 27 MCM. In the West Bank, the average annual potential RRWH volume is estimated at 37 MCM [11]. Therefore, the proper implementation of RRWH can potentially bridge the increasing domestic water supply-demand gap in the West Bank.

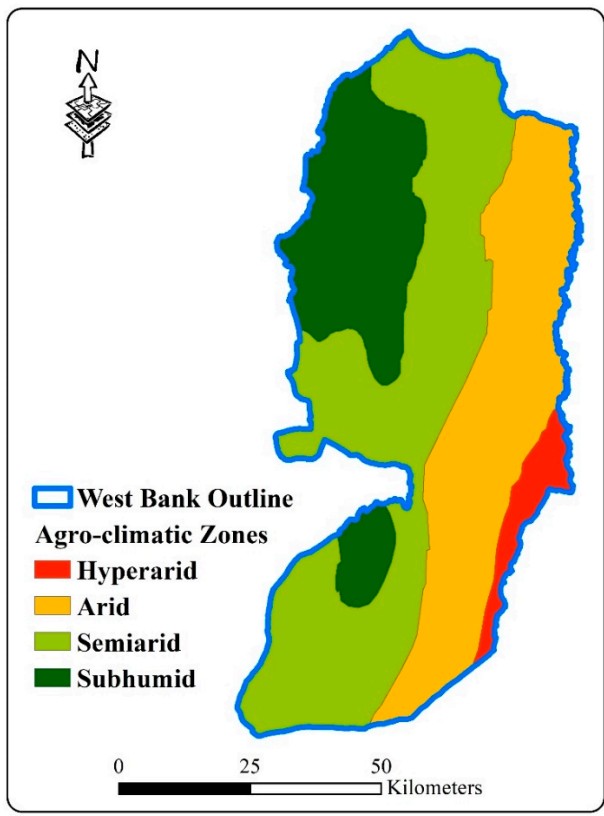

**Figure 2.** Agro-climatic zones in the West Bank.

*2.2. Methodology*

As mentioned earlier, there are different approaches to estimate the optimal RRWHo tank size. In this study, the required tank size (RRWHr) was estimated based on the continuous mass balance approach as presented by Rippl [27].

$$\Delta S = RRWH_V - DWD_V \qquad (1)$$

where: $\Delta S$ is the change in storage, $RRWH_V$ is the potential (monthly/daily) inflow volume to the storage tank, and $DWD_V$ is the (monthly/daily) outflow volume from the storage tank.

In this study, two RRWH scenarios are assumed and assessed. These are the annual, 12 months (S1), and the rainy, 8 months (S2). In these scenarios, the monthly time step was used in the estimation of RRWHo. The rainfall all-over the hydrologic year from October to September and over the rainy months from October to May was assumed for S1 and S2, respectively. However, and due to the lack of daily rainfall data, S1 and S2 scenarios were only tested given the available daily rainfall records at Nablus. DWDv is estimated based on the average DWD of 90 L/c/d and average family members of 4.8. As such, monthly $DWD_V$ of 13 m$^3$ (for all governorates) and daily $DWD_V$ of 0.43 m$^3$ (for Nablus) was used [34]. For S2, the estimated RRWHo tank size can be used to cover the DWD in the 8 rainy months. However, in the dry period from June to September, the tank can be utilized to store water from the municipal sources because, in most of the West Bank governorates, water is being supplied 2 days a week.

The $RRWH_V$ is estimated based on Equation (2) [41].

$$RRWH_V = R \times A \times R_C \qquad (2)$$

where: R is the average monthly/daily rainfall depth, A is the rooftop area, and $R_C$ is the runoff coefficient.

The average monthly (all governorates, Table 1) and daily (Nablus governorate, Table 2) rainfall data for some selected rainfall stations (presented in Figure 1) for the period (2013–2019) were obtained from the Palestinian Metrological Department [42].

**Table 1.** The average monthly rainfall (mm) in the different West Bank governorates for the period (2013–2019).

| Governorate | Jerusalem | Jenin | Tulkarm | Qalqiliya | Ramallah & Al-Bireh | Nablus | Bethlehem | Hebron | Jericho | Salfit | Tubas |
|---|---|---|---|---|---|---|---|---|---|---|---|
| Oct | 10 | 20 | 21 | 30 | 20 | 22 | 17 | 26 | 6 | 35 | 14 |
| Nov | 60 | 46 | 59 | 67 | 68 | 49 | 67 | 47 | 20 | 68 | 36 |
| Dec | 78 | 122 | 193 | 196 | 181 | 200 | 102 | 142 | 38 | 214 | 106 |
| Jan | 111 | 125 | 135 | 128 | 123 | 154 | 131 | 138 | 32 | 149 | 101 |
| Feb | 88 | 89 | 103 | 92 | 103 | 113 | 93 | 92 | 32 | 102 | 78 |
| Mar | 51 | 40 | 47 | 49 | 59 | 51 | 56 | 52 | 15 | 63 | 33 |
| Apr | 30 | 30 | 22 | 30 | 33 | 29 | 33 | 28 | 15 | 32 | 25 |
| May | 6 | 7 | 6 | 5 | 7 | 6 | 6 | 6 | 6 | 5 | 8 |
| Jun | 0 | 0 | 1 | 5 | 0 | 1 | 0 | 0 | 0 | 0 | 0 |
| Jul | 0 | 0 | 0 | 0 | 0 | 0 | 0 | 0 | 0 | 0 | 0 |
| Aug | 0 | 0 | 0 | 0 | 0 | 0 | 0 | 0 | 0 | 0 | 0 |
| Sep | 2 | 1 | 0 | 1 | 1 | 0 | 1 | 2 | 0 | 1 | 0 |

**Table 2.** Nablus daily rainfall for the period (2013–2019).

| Year | Sum (mm) | MADR * (mm) | Number of Rainy Days (Rainfall > 1 mm) |
|---|---|---|---|
| 2012/2013 | 701 | 107 | 44 |
| 2013/2014 | 466 | 123 | 24 |
| 2014/2015 | 667 | 83 | 53 |
| 2015/2016 | 498 | 48 | 42 |
| 2016/2017 | 476 | 74 | 41 |
| 2017/2018 | 566 | 61 | 42 |
| 2018/2019 | 925 | 90 | 59 |

MADR *: Maximum annual daily rainfall.

For the aforementioned scenarios, the RRWHr values were estimated based on the maximum cumulative sum of positive (monthly/daily) differences between DWDv and RRWHv. Whereas, the annual potential (maximum) tank size (RRWHm) is estimated based on the average annual rainfall values (2013–2019) of the selected rainfall stations at different governorates.

Moreover, rooftop areas of 100 m$^2$, 150 m$^2$, 200 m$^2$, 250 m$^2$, and 300 m$^2$ were used in the estimation of RRWH$_V$ values. R$_C$ of 0.8 was selected, which is in the range of R$_C$ values that appeared in the body of literature for the concrete rooftops [43–46]. RRWHo values for each governorate and the different rooftop areas are set as the minimum of either RRWHr or RRWHm values.

The reliability (Re) of adopting an RRWH system in the West Bank was tested based on Equation (3) [47].

$$Re = RRWH_V / DWD_V \tag{3}$$

The overall methodological approach for this research is illustrated in Figure 3.

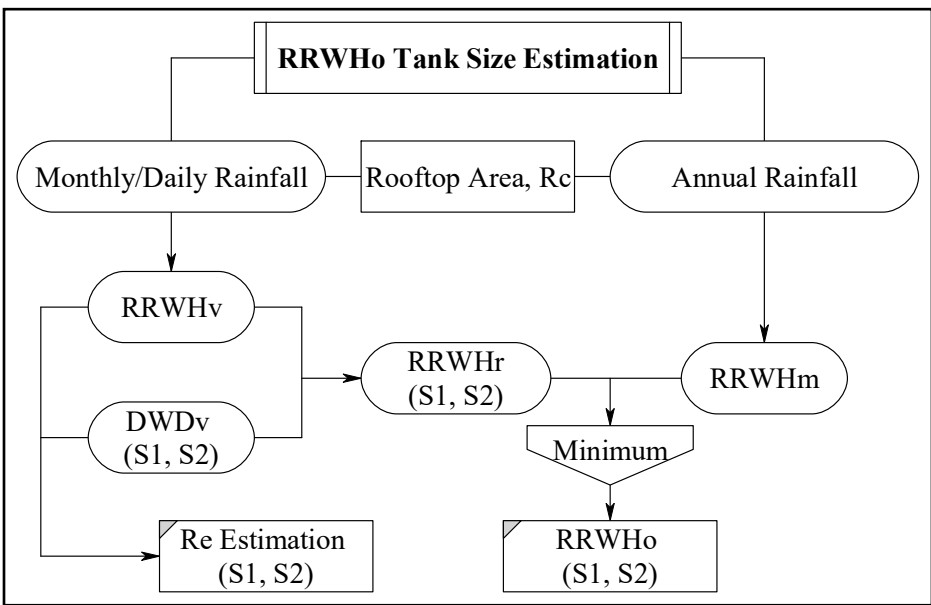

**Figure 3.** The overall methodological approach.

## 3. Results and Discussion

### 3.1. RRWHo Tank Size Estimation

Based on the available rainfall data and given the aforementioned approach, $RRWH_O$ values for the different governorates were estimated. For Nablus, as an example, and given the two scenarios, calculations of RRWHr, RRWHm, and RRWHo for the different rooftop areas are presented in Tables 3 and 4 for S1 and S2 scenarios, respectively. For all governorates, calculations of RRWHr, RRWHm, and RRWHo are presented in the Supplementary Materials.

From Tables 3 and 4, and to satisfy the DWD of 90 L/c/d for an average family size of 4.8 members, it is noticed that the RRWHr is inversely proportioned with the rooftop areas for the two scenarios, while the RRWHm is being increased as the rooftop area increased. For the S1 scenario and rooftop areas of 150 m$^2$ and less, the RRWHo tank size is being controlled by the RRWHm. Whereas, for rooftop areas of 200 m$^2$ and more, RRWHo tank size is being controlled by the RRWHr. For rooftop areas of less than 100 m$^2$, the RRWHm represents the RRWHo, and for the rooftop areas of more than 150 m$^2$, the RRWHo values are being controlled by the RRWHr values and for S1 and S2 scenarios.

Table 5 illustrates the RRWHo tank sizes for Nablus governorate for the two scenarios given the monthly and daily rainfall data. It is clear from the table that the RRWHo values obtained from both monthly and daily rainfall data are almost the same. This in turn supports the use of monthly rainfall data for the estimation of RRWHo tank size.

**Table 3.** RRWHo calculations for Nablus governorate for the different rooftop areas for the S1 scenario.

| Month | Rainfall (mm) | 100 | | | | 150 | | | | 200 | | | | 250 | | | | 300 | | | |
|---|---|---|---|---|---|---|---|---|---|---|---|---|---|---|---|---|---|---|---|---|---|
| | | 1 | 2 | 3 | 4 | 1 | 2 | 3 | 4 | 1 | 2 | 3 | 4 | 1 | 2 | 3 | 4 | 1 | 2 | 3 | 4 |
| Oct | 21.8 | 1.7 | 13.0 | 11.2 | 11.2 | 2.6 | 13.0 | 10.3 | 10.3 | 3.5 | 13.0 | 9.5 | 9.5 | 4.4 | 13.0 | 8.6 | 8.6 | 5.2 | 13.0 | 7.7 | 7.7 |
| Nov | 49.4 | 4.0 | 13.0 | 9.0 | 20.2 | 5.9 | 13.0 | 7.0 | 17.4 | 7.9 | 13.0 | 5.1 | 14.5 | 9.9 | 13.0 | 3.1 | 11.7 | 11.9 | 13.0 | 1.1 | 8.8 |
| Dec | 199.8 | 16.0 | 13.0 | −3.0 | 17.2 | 24.0 | 13.0 | −11.0 | 6.4 | 32.0 | 13.0 | −19.0 | −4.5 | 40.0 | 13.0 | −27.0 | −15.3 | 48.0 | 13.0 | −35.0 | −26.2 |
| Jan | 154.1 | 12.3 | 13.0 | 0.6 | 17.8 | 18.5 | 13.0 | −5.5 | 0.8 | 24.7 | 13.0 | −11.7 | −16.2 | 30.8 | 13.0 | −17.9 | −33.2 | 37.0 | 13.0 | −24.0 | −50.2 |
| Feb | 112.5 | 9.0 | 13.0 | 4.0 | 21.8 | 13.5 | 13.0 | −0.54 | 0.3 | 18.0 | 13.0 | −5.0 | −21.2 | 22.5 | 13.0 | −9.54 | −42.7 | 27.0 | 13.0 | −14.0 | −64.2 |
| Mar | 51.0 | 4.1 | 13.0 | 8.9 | 30.7 | 6.1 | 13.0 | 6.8 | 7.1 | 8.2 | 13.0 | 4.8 | −16.4 | 10.2 | 13.0 | 2.8 | −40.0 | 12.2 | 13.0 | 0.7 | −63.5 |
| Apr | 29.3 | 2.3 | 13.0 | 10.6 | 41.3 | 3.5 | 13.0 | 9.4 | 16.6 | 4.7 | 13.0 | 8.3 | −8.1 | 5.9 | 13.0 | 7.1 | −32.9 | 7.0 | 13.0 | 5.9 | −57.6 |
| May | 6.4 | 0.5 | 13.0 | 12.4 | 53.7 | 0.8 | 13.0 | 12.2 | 28.8 | 1.0 | 13.0 | 11.9 | 3.8 | 1.3 | 13.0 | 11.7 | −21.2 | 1.5 | 13.0 | 11.4 | −46.2 |
| Jun | 0.6 | 0.0 | 13.0 | 12.9 | 66.6 | 0.1 | 13.0 | 12.9 | 41.7 | 0.1 | 13.0 | 12.9 | 16.7 | 0.1 | 13.0 | 12.8 | −8.3 | 0.1 | 13.0 | 12.8 | −33.3 |
| Jul | 0.0 | 0.0 | 13.0 | 13.0 | 79.6 | 0.0 | 13.0 | 13.0 | 54.6 | 0.0 | 13.0 | 13.0 | 29.6 | 0.0 | 13.0 | 13.0 | 4.6 | 0.0 | 13.0 | 13.0 | −20.4 |
| Aug | 0.0 | 0.0 | 13.0 | 13.0 | 92.6 | 0.0 | 13.0 | 13.0 | 67.6 | 0.0 | 13.0 | 13.0 | 42.6 | 0.0 | 13.0 | 13.0 | 17.6 | 0.0 | 13.0 | 13.0 | −7.4 |
| Sep | 0.1 | 0.0 | 13.0 | 13.0 | 105.5 | 0.0 | 13.0 | 12.9 | 80.5 | 0.0 | 13.0 | 12.9 | 55.5 | 0.0 | 13.0 | 12.9 | 30.5 | 0.0 | 13.0 | 12.9 | 5.5 |
| Annual | 625.0 | 50.0 | 155.5 | | | 75.0 | 155.5 | | | 100.0 | 155.5 | | | 125.0 | 155.5 | | | 150.0 | 155.5 | | |
| RRWHr | | | | | 106 | | | | 81 | | | | 56 | | | | 31 | | | | 9 |
| RRWHm | | | | | 50 | | | | 75 | | | | 100 | | | | 125 | | | | 150 |
| RRWHo | | | | | 50 | | | | 75 | | | | 56 | | | | 31 | | | | 9 |
| Rv (%) | | | | | 32 | | | | 48 | | | | 64 | | | | 80 | | | | 96 |

1 = RRWHv, 2 = DWD$_V$, 3 = DWD$_V$ − RRWHv, 4 = (DWD$_V$ − RRWHv) cumulative.

**Table 4.** RRWHo calculations for Nablus governorate for the different rooftop areas for the S2 scenario.

| Month | Rainfall (mm) | 100 | | | | 150 | | | | 200 | | | | 250 | | | | 300 | | | |
|---|---|---|---|---|---|---|---|---|---|---|---|---|---|---|---|---|---|---|---|---|---|
| | | 1 | 2 | 3 | 4 | 1 | 2 | 3 | 4 | 1 | 2 | 3 | 4 | 1 | 2 | 3 | 4 | 1 | 2 | 3 | 4 |
| Oct | 21.8 | 1.7 | 13.0 | 11.2 | 11.2 | 2.6 | 13.0 | 10.3 | 10.3 | 3.5 | 13.0 | 9.5 | 9.5 | 4.4 | 13.0 | 8.6 | 8.6 | 5.2 | 13.0 | 7.7 | 7.7 |
| Nov | 49.4 | 4.0 | 13.0 | 9.0 | 20.2 | 5.9 | 13.0 | 7.0 | 17.4 | 7.9 | 13.0 | 5.1 | 14.5 | 9.9 | 13.0 | 3.1 | 11.7 | 11.9 | 13.0 | 1.1 | 8.8 |
| Dec | 199.8 | 16.0 | 13.0 | −3.0 | 17.2 | 24.0 | 13.0 | −11.0 | 6.4 | 32.0 | 13.0 | −19.0 | −4.5 | 40.0 | 13.0 | −27.0 | −15.3 | 48.0 | 13.0 | −35.0 | −26.2 |
| Jan | 154.1 | 12.3 | 13.0 | 0.6 | 17.8 | 18.5 | 13.0 | −5.5 | 0.8 | 24.7 | 13.0 | −11.7 | −16.2 | 30.8 | 13.0 | −17.9 | −33.2 | 37.0 | 13.0 | −24.0 | −50.2 |
| Feb | 112.5 | 9.0 | 13.0 | 4.0 | 21.8 | 13.5 | 13.0 | −0.5 | 0.3 | 18.0 | 13.0 | −5.0 | −21.2 | 22.5 | 13.0 | −9.5 | −42.7 | 27.0 | 13.0 | −14.0 | −64.2 |
| Mar | 51.0 | 4.1 | 13.0 | 8.9 | 30.7 | 6.1 | 13.0 | 6.8 | 7.1 | 8.2 | 13.0 | 4.8 | −16.4 | 10.2 | 13.0 | 2.8 | −40.0 | 12.2 | 13.0 | 0.7 | −63.5 |
| Apr | 29.3 | 2.3 | 13.0 | 10.6 | 41.3 | 3.5 | 13.0 | 9.4 | 16.6 | 4.7 | 13.0 | 8.3 | −8.1 | 5.9 | 13.0 | 7.1 | −32.9 | 7.0 | 13.0 | 5.9 | −57.6 |
| May | 6.4 | 0.5 | 13.0 | 12.4 | 53.7 | 0.8 | 13.0 | 12.2 | 28.8 | 1.0 | 13.0 | 11.9 | 3.8 | 1.3 | 13.0 | 11.7 | −21.2 | 1.5 | 13.0 | 11.4 | −46.2 |
| Annual | 624.3 | 49.9 | 103.7 | | | 74.9 | 103.7 | | | 99.9 | 103.7 | | | 124.9 | 103.7 | | | 149.8 | 103.7 | | |
| RRWHr | | | | | 54 | | | | 29 | | | | 15 | | | | 12 | | | | 9 |
| RRWHm | | | | | 50 | | | | 75 | | | | 100 | | | | 125 | | | | 150 |
| RRWHo | | | | | 50 | | | | 29 | | | | 15 | | | | 12 | | | | 9 |
| Rv (%) | | | | | 48 | | | | 72 | | | | 96 | | | | 120 | | | | 145 |

1 = RRWHv, 2 = DWD$_V$, 3 = DWD$_V$ − RRWHv, 4 = (DWD$_V$ − RRWHv) cumulative.

**Table 5.** Comparison between RRWHo values for Nablus governorate for the two scenarios and based on the monthly and daily rainfall data.

| Rooftop Areas (m²) | RRWHo Tank Size (m³) | | | |
|---|---|---|---|---|
| | S1 | | S2 | |
| | Monthly | Daily | Monthly | Daily |
| 100 | 50 | 49 | 50 | 49 |
| 150 | 75 | 75 | 29 | 31 |
| 200 | 56 | 59 | 15 | 15 |
| 250 | 31 | 35 | 12 | 13 |
| 300 | 9 | 10 | 9 | 11 |

In Figure 4, the RRWHr for the two scenarios and the RRWHm versus rooftop areas for the different governorates are shown.

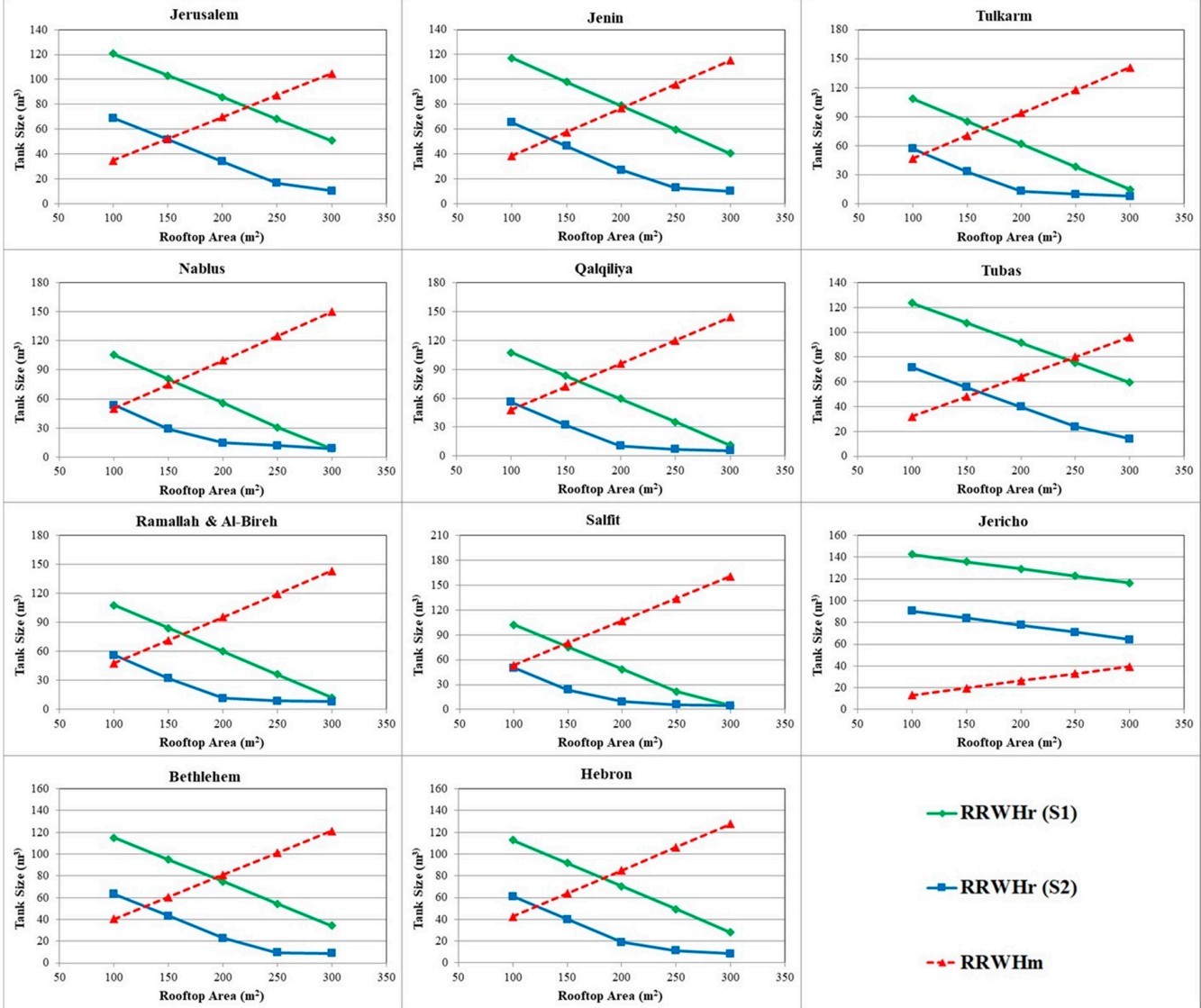

**Figure 4.** RRWHr and RRWHm versus different rooftop areas in the different governorates for S1 and S2 scenarios.

From Figure 4, it is noticed that the RRWHo values for S1 (annual) are more than the RRWHo values for S2 (rainy). This can be attributed to the highly cumulative differences between $DWD_V$ and $RRWH_V$. For the S1 scenario, in the governorates of Tulkarm, Nablus,

Qalqiliya, Hebron, and Ramallah and Al-Bireh, the RRWHr tank size exceeds the RRWHm tank size for the rooftop areas of less than 150 m$^2$ (RRWHo equals RRWHm). However, RRWHr tank sizes are less than the RRWHm tank size in Jenin and Bethlehem for the rooftop areas of more than 200 m$^2$ (RRWHo equals RRWHr).

In governorates of Nablus, Qalqiliya, Salfit, and Ramallah and Al-Bireh, the RRWHr values are controlling the RRWHo tank sizes for the S2 scenario and rooftop areas of more than 100 m$^2$. In Jericho (the least rainfall governorate), the RRWHm are dominating the selection of RRWHo tank sizes for S1 and S2 scenarios and the different rooftop areas. That means satisfying DWD in Jericho is not feasible by adopting the RRWH techniques either for annual (12 months) or rainy (8 months).

For the average rooftop area of 150 m$^2$, the spatial variation of RRWHo tank sizes for the different governorates for S1 and S2 are presented in Figure 5. From the figure, it is clear that the minimum value of RRWHo (20 m$^3$, in Jericho) and for S1 and S2 scenarios. Whereas, the maximum value of RRWHo (75 m$^3$, in Salfit and Nablus) and (51 m$^3$, in Jerusalem) for S1 and S2 scenarios, respectively.

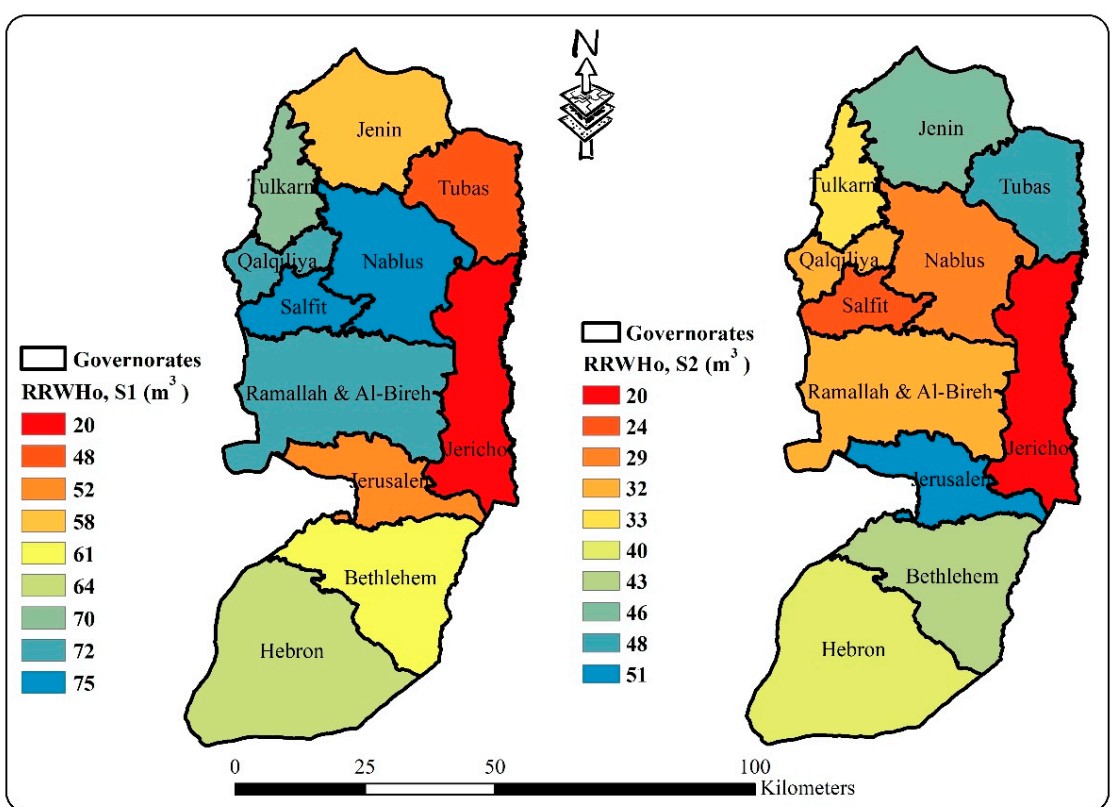

**Figure 5.** RRWHo tank size for a rooftop area of 150 m$^2$ for the different West Bank governorates and the S1 and S2 scenarios.

### 3.2. Reliability of RRWH

In this research, the reliability (Re) of adopting RRWH was tested for S1 and S2 scenarios at the different rooftop areas (100 m$^2$, 150 m$^2$, 200 m$^2$, 250 m$^2$, and 300 m$^2$) and average monthly DWDv (13 m$^3$/family).

Figure 6 presents the variation of Re for the S1 and S2 and different rooftop areas. Generally, the Re values increase as the rooftop area increases. This can be attributed to the increased values of the estimated RRWH$_V$. It is also noticed that for a rooftop area of 150 m$^2$, the maximum value of Re is in Salfit governorate and the minimum one in Jericho for both scenarios. Accordingly, Re in Salfit is 52% and 77% for S1 and S2, respectively. While in Jericho, Re equals 13% for S1 and 19% for S2.

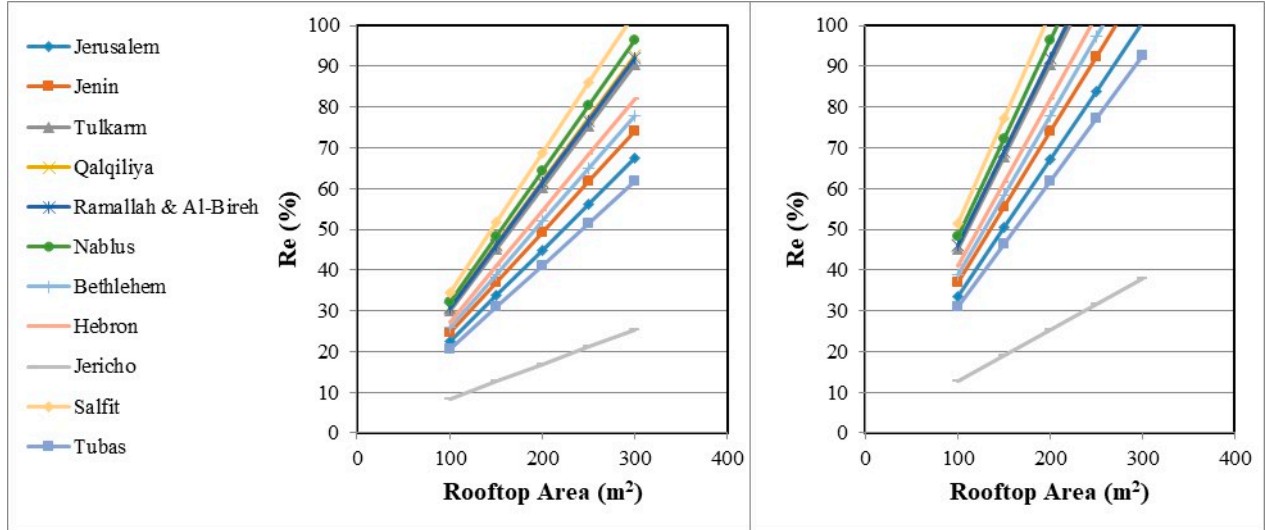

**Figure 6.** Re percentages of different West Bank governorates for the different rooftop areas for a monthly DWDv of 13 m³/family. S1 (**left**); S2 (**right**).

Moreover, the Re was tested for a monthly $DWD_V$ of 6.5 m³/family (50% of the average DWD in the West Bank) at the different rooftop areas (see Figure 7). From the figure, generally, Re increased as DWD decreased. For example, at a rooftop area of 150 m², the Re values for Salfit increased to nearly 100% for S1 and S2. Meanwhile, in Jericho (the least reliable governorate), Re increased from 13% to 25% for S1 and from 19% to 38% for S2. Additionally, the Re of adopting RRWH increased to more than 80% for the rooftop areas of 200 m² and more for all of the West Bank governorates except for Jericho. Therefore, to assess the effect of reducing DWD on the reliability of adopting an RRWH system in Jericho, family $DWD_V$ values of 3.2, 1.3, and 1 m³/month, which represent 25%, 10%, and 8% from the average family DWDv in the West Bank (13 m³/month), respectively (see Table 6). From the table, Re of 100% can be obtained at rooftop areas of 100 m² and more if DWDv decreased to 8% and 10% for the S1 and S2 scenarios, respectively. At a 150 m² rooftop area, Re values approach 100% for a family DWDv of 1.3 m³/month (10% of the average $DWD_V$) for both S1 and S2 scenarios.

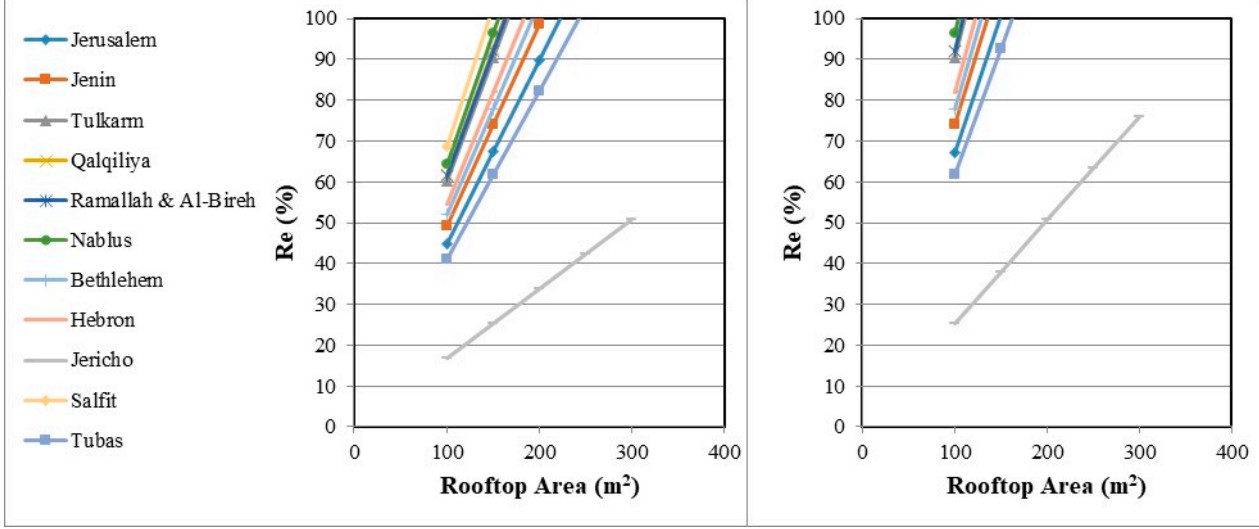

**Figure 7.** Re percentages of different West Bank governorates for the different rooftop areas for a monthly DWDv of 6.5 m³/family. S1 (**left**); S2 (**right**).

**Table 6.** Re values for selected $DWD_V$ values (percentages from the average $DWD_V$) in Jericho governorate.

| Rooftop Areas (m$^2$) | Re (%) | | | | | |
|---|---|---|---|---|---|---|
| | S1 | | | S2 | | |
| | 25% | 10% | 8% | 25% | 10% | 8% |
| 100 | 34 | 85 | - | 51 | - | - |
| 150 | 51 | - | - | 76 | - | - |
| 200 | 68 | - | - | - | - | - |
| 250 | 85 | - | - | - | - | - |
| 300 | - | - | - | - | - | - |

"-" Re values more than 99%.

## 4. Conclusions

This paper presents a realistic estimation of RRWHo tank size in the West Bank based on the continuous mass balance (Rippl) approach. Moreover, the reliability of the implementation of RRWH techniques in the different governorates was also tested. The estimation of RRWHo tank size was accomplished based on the average monthly DWDv of 13 m$^3$/family and based on two scenarios (S1, annual, and S2, rainy). For the average rooftop area (150 m$^2$), results indicate that the maximum RRWHo tank size (75 m$^3$) is in Salfit and Nablus and 51 m$^3$ in Jerusalem for S1 and S2, respectively. The minimum one is in Jericho (20 m$^3$) for both scenarios. Reliability results show that the implementation of RRWH techniques is mostly reliable (S1, Re = 52% and S2, Re = 77%) in Salfit to satisfy the average DWD of 90 L/c/d at a rooftop area of 150 m$^2$ and a family size of 4.8 members. The reliability of adopting RRWH is inversely proportional to the decrease of DWDv. Moreover, in Jericho governorate (rainfall is the least in the West Bank, 133 mm/year), Re of about 100% can be achieved at a rooftop area of 150 m$^2$ and a reduction of a family $DWD_V$ to 10% (1.3 m$^3$/month) for S1 and S2 scenarios.

The developed RRWHr tank size curves for the different West Bank governorates will help different stakeholders in the designing of optimal RRWH storage tanks for the different rooftop areas to fulfill domestic water needs, totally or partially. Whereas, the reliability assessment will guide decision-makers toward sustainable implementation of RRWH techniques in the different West Bank governorates for the different rooftop areas. Finally, further research is recommended to incorporate social and economic factors for designing optimal RRWH storage tanks to promote the utilization of RRWH for a self-sustaining and self-reliant water supply in Palestine.

**Supplementary Materials:** The following are available online at https://www.mdpi.com/2073-4441/13/4/573/s1.

**Author Contributions:** Conceptualization, S.S.; methodology, S.S. and S.A.; GIS, S.A., and S.S.; formal analysis, S.S.; writing—original draft preparation, S.A.; writing—review and editing of the final version, S.S.; project administration, S.S. All authors have read and agreed to the published version of the manuscript.

**Funding:** This research was performed within the framework of the Palestinian Dutch Academic Cooperation Program on Water (PADUCO 2), funded by the Netherlands Representative Office (NRO) in Ramallah, Palestine. The financial support is gratefully acknowledged.

**Institutional Review Board Statement:** Not applicable.

**Informed Consent Statement:** Not applicable.

**Data Availability Statement:** The data presented in this study are available in the Supplementary Materials.

**Conflicts of Interest:** The authors declare no conflict of interest.

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
