# Peer review of "Optimal Sizing of Rooftop Rainwater Harvesting Tanks for Sustainable Domestic Water Use in the West Bank, Palestine"

_water, doi:10.3390/w13040573_

Round 1

Reviewer 1 Report

The authors have addressed the major comments made earlier. The manuscript is acceptable after following minor corrections:

  1. Lines 77-80: The authors cited several works from different countries. A significant number of works done in Australia, however authors did not cite any of those papers. Please see some papers from Imteaz et al. and Rahman et al. for Australia.
  2. Uses of English are incorrect in many places. A thorough correction of English is required. A few examples are: (i) Line 69 "The estimation of RRWH tank size can be estimated", estimation used twice.(ii) Line 81: "This research study aims at estimating of optimal RRWH". (iii) Line 89: "The specific novel of this research.." (iv) Line 141: "However, and due the limited available daily rainfall.."

Author Response

Once again, we would like to thank you for your effort and positive feedback. All your comments are valid and appreciated and with no doubt enhanced the quality of the manuscript. However, all of them were considered in the revised version (see the track changes).

Reviewer 2 Report

Dear Authors,

I really appreciated your efforts for the improvements of the manuscript.

My requests of revision were satisfied. 

The quality of presentation is good, but the originality and novelty of the work is limited, as well as its scientific soundness. Contents are significant to guide local operators and designers since the analysis is thorough and interesting for the specific context.

I only have few more observations:

  • I suggest to further increase the bibliography review.
  • I have some doubts about how the obtained results could be mirrored in any country of the world, so I ask to better explain this sentence at the end of the introduction.
  • Better explain differences between Fig 6 and Fig 7 (in Figure 6 DWD is not specified).

Author Response

Once again, we would like to thank you for your effort and positive feedback. All your comments are valid and appreciated and with no doubt enhanced the quality of the manuscript. However, all of them were considered in the revised version (see the track changes).

This manuscript is a resubmission of an earlier submission. The following is a list of the peer review reports and author responses from that submission.

Round 1

Reviewer 1 Report

Although the manuscript covered a wide area, however the method applied is not acceptable for scientific publishing. Researches in this area has gone very far, many researchers (Imteaz et al., Rahman et al., Ghisi et al. and more) used daily water balance model for such studies and they have published numerous papers on this. A daily scale model is good enough to produce accurate results, although some researchers even started using hourly scale model to produce more accurate results. The authors used monthly scale model, which is of very primary level (even I ask first-year engineering students to do assignment using monthly scale model). Such monthly scale model can not produce accurate results, as it can not replicate overflow losses, which are obvious during consecutive rainy days. For the case of study area, the period between December-February are having very high rainfalls, when the tank is likely to overflow. In monthly scale model, this is not possible to take into consideration. This is the major drawback of the study, which does not qualify the manuscript to be published in a good-quality journal. As such, I reject the paper. 

Reviewer 2 Report

Dear Authors,

I think that the manuscript could be suitable for publication on Water, but I observed three main weakness for its publication.

  • The originality/novelty of the paper is not clear.
  • Proposed method to estimate RRWH tank size is not compared with other methods suggested in literature
  • The research has a local interest since results cannot be generalized

For this aspect, I require to you a major revision of the manuscript. I ask to better define the contribution of the paper to research in the field of RRWH and to compare obtained results with other design method proposed in literature.

In the following I reported specific suggestions for each section.

“Introduction”

It can be improved.  I suggest extending the bibliographic review of design methods (lines 53-58) considering also semi-probabilistic approach to estimate RRWH tank size. I would better present also the analysis carried out in the manuscript (lines 59-63). In particular, the method used in the estimation of the RRWH tank size is not defined, as well as the originality of the study. I suggest to clearly state the contribution of the paper to research in the field of RRWH. The added values declared in lines 64-65 seems too weak because I did not find reliable relation between rooftop areas and optimal RRWH storage tank size in the text but only trend depending on local set data specific of the case study.

“2.1 Study area”

Lines 75-76: it would be interesting to show (e.g., in a map as that shown in Figure 1), where the four climatic zones are located.

Line 79-81: I would add a reference to support this sentence (“In this research, the focus is on the built-up areas where rooftops account for about 28% that was considered in the estimation of RRWH volumes in the different governorates”).

“2.2 Methodology”

Equation (2): please put v as subscript in agree with equation (1).

Figure 2: the scheme of the methodological approach is not clear. Please improve it.

“3.1. RRWHo Tank Size Estimation”

Table 2: I think there is a mistake: the sentence: “1= RRWHv, 2= RRWHv-DWD1, 3= RRWHv-DWD2” in legend, doesn’t have correspondence in the table (I think the mistake is (2,3,4) instead of (1,2,3) in the second line of the table.

Please move Table 2 after line 137.

Lines 138-143: better justify obtained results.

Please put Table 3 after line 176.

“Conclusions”

Line 195: review the syntax of this sentence.

Reviewer 3 Report

I enjoyed reading this manuscript. Just like many other theoretical approaches for rainwater harvesting systems, it is really like a beautiful fairytale.

The problem is that when it comes to real-scaled implementation, things are much different from our dreams on pieces of paper.

The reviewer of this work has practiced rainwater harvesting systems on different scales. It is indeed one of my favorites, and I do not deny the potential merits of the idea of RRWH. But can it be the main source of domestic water? Or is it an alternative one?

I appreciate the author's approach. But considering the information presented in table 1, I doubt that the system can be sustainable for the mentioned area.

Here are more specific comments on the manuscript:

1. Authors are misunderstood about using reliability (Re) as stated in equation 3. Re is the reliability as the fraction of time that the demand is fully met. This means that the Re-value is valid only and only when the storage capacity exceeds a certain volume. Therefore using Re as formulated in equation 3 can be valid only when 1. there is enough amount of precipitation (you have enough rain to fill your rainwater tank), 2. You have notably large storage capacity, and 3. water demand is low. Based on the information presented in table 2, and the mentioned assumptions, I do not think that using Re can be scientifically correct, and therefore, the results cannot valid due to the incorrect methodology.

2. The whole paper is a desktop work, that's why I say it is as beautiful as a fairytale. There are serious challenges when it comes to real-scaled implementation. In my point of view designing the tank size based on the demand, before spillage, without sufficient consideration of the dry period of the year, overestimates the calculation. Otherwise, authors should clearly mention that their objective of implementing an RRWH system is providing water for a partial amount of demand, which means that the system is actually an alternative water resource.

3. The author should discuss their strategy for dry seasons (when rainfall is below 60mm). This period is almost 9 months in the studied areas.

Overall, please note that one of the bottlenecks of rainwater harvesting all over the world is the seasonality of rainfall and its non-availability during the dry season and drought periods. Solutions to this include better forecasting and acquisition of adequately sized cistern. To provide an adequately sized cistern we do not have to consider meeting all the water demands of a family. Even if we can cover 5%-10% of the water demand, it could definitely be a robust approach. Also, real-scaled experiments and demonstrations are essential before claiming an approach as a "realistic" one.